# Prognostic Role of Survivin and Macrophage Infiltration Quantified on Protein and mRNA Level in Molecular Subtypes Determined by RT-qPCR of *KRT5*, *KRT20*, and *ERBB2* in Muscle-Invasive Bladder Cancer Treated by Adjuvant Chemotherapy

**DOI:** 10.3390/ijms21197420

**Published:** 2020-10-08

**Authors:** Thorsten H. Ecke, Adisch Kiani, Thorsten Schlomm, Frank Friedersdorff, Anja Rabien, Klaus Jung, Ergin Kilic, Peter Boström, Minna Tervahartiala, Pekka Taimen, Jan Gleichenhagen, Georg Johnen, Thomas Brüning, Stefan Koch, Jenny Roggisch, Ralph M. Wirtz

**Affiliations:** 1Department of Urology, HELIOS Hospital Bad Saarow, DE-15526 Bad Sarrow, Germany; 2Brandenburg Medical School, DE-14770 Brandenburg, Germany; stefan.koch@helios-gesundheit.de; 3Department of Urology, Charité—Universitätsmedizin, Corporate Member of Freie Universität Berlin, Humboldt-Universität zu Berlin, and Berlin Institute of Health, DE-10098 Berlin, Germany; adisch.kiani@charite.de (A.K.); thorsten.schlomm@charite.de (T.S.); frank.friedersdorff@charite.de (F.F.); Anja.Rabien@charite.de (A.R.); klaus.jung@charite.de (K.J.); 4Berlin Institute for Urological Research, DE-10098 Berlin, Germany; 5Institute of Pathology, DE-51375 Leverkusen, Germany; e.kilic@pathologie-leverkusen.de; 6Department of Urology, Turku University Hospital, FI-20521 Turku, Finland; peter.j.bostrom@gmail.com; 7MediCity Research Laboratory, Department of Medical Microbiology and Immunology, University of Turku, FI-20520 Turku, Finland; mmbost@utu.fi; 8Institute of Pathology, Turku University Hospital, FI-20521 Turku, Finland; pepeta@utu.fi; 9Institute for Prevention and Occupational Medicine of the German Social Accident Insurance (IPA), Institute of the Ruhr University Bochum, DE-44789 Bochum, Germany; Gleichenhagen@ipa-dguv.de (J.G.); johnen@ipa-dguv.de (G.J.); bruening@ipa-dguv.de (T.B.); 10Institute of Pathology, HELIOS Hospital Bad Saarow, DE-15526 Bad Sarrow, Germany; jenny.roggisch@helios-gesundheit.de; 11STRATIFYER Molecular Pathology GmbH, DE-50935 Cologne, Germany; ralph.wirtz@stratifyer.de

**Keywords:** survivin, BIRC5, macrophage, KRT20, ERBB2, MIBC, prediction, RT-qPCR, adjuvant chemotherapy, survival, bladder cancer

## Abstract

Objectives: Bladder cancer is a heterogeneous malignancy. Therefore, it is difficult to find single predictive markers. Moreover, most studies focus on either the immunohistochemical or molecular assessment of tumor tissues by next-generation sequencing (NGS) or PCR, while a combination of immunohistochemistry (IHC) and PCR for tumor marker assessment might have the strongest impact to predict outcome and select optimal therapies in real-world application. We investigated the role of proliferation survivin/*BIRC5* and macrophage infiltration (CD68, MAC387, CLEVER-1) on the basis of molecular subtypes of bladder cancer (KRT5, KRT20, ERBB2) to predict outcomes of adjuvant treated muscle-invasive bladder cancer patients with regard to progression-free survival (PFS) and disease-specific survival (DSS). Materials and Methods: We used tissue microarrays (TMA) from n = 50 patients (38 males, 12 female) with muscle-invasive bladder cancer. All patients had been treated with radical cystectomy followed by adjuvant triple chemotherapy. Median follow-up time was 60.5 months. CD68, CLEVER-1, MAC387, and survivin protein were detected by immunostaining and subsequent visual inspection. *BIRC5*, *KRT5*, *KRT20*, *ERBB2*, and *CD68* mRNAs were detected by standardized RT-qPCR after tissue dot RNA extraction using a novel stamp technology. All these markers were evaluated in three different centers of excellence. Results: Nuclear staining rather than cytoplasmic staining of survivin predicted DSS as a single marker with high levels of survivin being associated with better PFS and DSS upon adjuvant chemotherapy (*p* = 0.0138 and *p* = 0.001, respectively). These results were validated by the quantitation of *BIRC5* mRNA by PCR (*p* = 0.0004 and *p* = 0.0508, respectively). Interestingly, nuclear staining of survivin protein was positively associated with *BIRC5* mRNA, while cytoplasmic staining was inversely related, indicating that the translocation of survivin protein into the nucleus occurred at a discrete, higher level of its mRNA. Combining survivin/*BIRC5* levels based on molecular subtype being assessed by *KRT20* expression improved the predictive value, with tumors having low survivin/*BIRC5* and *KRT20* mRNA levels having the best survival (75% vs. 20% vs. 10% 5-year DSS, *p* = 0.0005), and these values were independent of grading, node status, and tumor stage in multivariate analysis (*p* = 0.0167). Macrophage infiltration dominated in basal tumors and was inversely related with the luminal subtype marker gene expression. The presence of macrophages in survivin-positive or *ERBB2*-positive tumors was associated with worse DSS. Conclusions: For muscle-invasive bladder cancer patients, the proliferative activity as determined by the nuclear staining of survivin or RT-qPCR on the basis of molecular subtype characteristics outperforms single marker detections and single technology approaches. Infiltration by macrophages detected by IHC or PCR is associated with worse outcome in defined subsets of tumors. The limitations of this study are the retrospective nature and the limited number of patients. However, the number of molecular markers has been restricted and based on predefined assumptions, which resulted in the dissection of muscle-invasive disease into tumor–biological axes of high prognostic relevance, which warrant further investigation and validation.

## 1. Introduction

Bladder cancer is the fifth most frequent cancer in Europe. In 2018, its incidence and annual mortality rate were estimated to reach 197,105 and 64,966 cases, respectively [1]. Approximately 30% of these patients suffered from muscle-invasive bladder cancer (MIBC) at the time of initial diagnosis [2]. Radical cystectomy (RC) is the gold standard to treat these patients. Compared to patients with non-muscle-invasive bladder cancer (NMIBC), MIBC patients are subject to a high risk of cancer-related death.

In order to remedy this unsatisfactory situation, serious efforts have recently focused on new therapeutic strategies regarding the application of neoadjuvant and adjuvant chemotherapies [3]. A better risk assessment of patients has been recommended by developing novel predictive/prognostic models [4]. In clinical practice, the therapeutic management of these patients has so far been performed almost exclusively on the basis of clinical data and classical pathological TNM criteria but with few reliable results [4]. It is hoped that the identification of new molecular tissue biomarkers could help to stratify risk groups and determine patients who could have a benefit from adjuvant strategies after surgery [5]. In the last decades, many different markers (nucleic acid or protein based) have been identified to add more information on risk assessment. A subset of different markers was selected and further investigated in this study.

Survivin, also known as baculoviral IAP repeat containing 5 (BIRC5), is a member of the inhibitor of apoptosis family and has an important role in cell cycle regulation [6]. The protein survivin is present in different tumor tissues; it occurs in cytoplasm, but also in nuclei [7,8]. The protein is very rarely present in normal tissue [9]. Survivin acts as an apoptosis suppressor in cytoplasm and nuclei and influences cell division [8]. If the stress signal is high enough, survivin is released into cytoplasm, which leads to the inhibition of different caspases [10]. The relevance of *BIRC5* mRNA expression has been less studied in detail. However, its high prognostic impact for certain bladder cancer stages has been shown in the prospective UROMOL trial, wherein it belongs to a 12 gene signature with an adverse effect on survival for NMIBC [11].

CD68 is the most frequently used pan-macrophage marker. Its function is still unknown, but it has been considered to play a role in the phagocytic activities of tissue macrophages [12]. Common lymphatic endothelial and vascular endothelial receptor-1 (CLEVER-1, also known as stabilin-1 or STAB1) is a multifunctional immunosuppressive scavenger receptor expressed by lymphatic and vascular endothelial cells and tissue macrophages [13]. Its prognostic significance in bladder cancer is not clear, but there is evidence that high CLEVER-1-positive macrophage count associates with chemoresistance [14] in neoadjuvant-treated bladder cancer patients. The monoclonal antibody MAC387 detects an epitope on the calcium-binding protein MRP14/S100A9 present in the cytosol of monocytes and granulocytes [15]. It is the exclusive arachidonic acid-binding protein in human neutrophils and is thereby involved in the calcium-dependent cellular signal of lipid second messengers during inflammatory and metabolic changes of tumor-associated macrophages [16].

The molecular subtyping of bladder cancer has been well accepted after its initial introduction in 2014 [17,18,19]. Therein, the quantitation of *KRT5* and *KRT20* on mRNA level and/or their recapitulation on protein level by immunohistochemistry (IHC) have been identified as exemplary biomarkers for the molecular subtyping of basal and luminal tumors, respectively. In our previous work, we could show that *KRT20* is strongly associated with adverse outcome for pT1 NMIBC [20].

*ERBB2* belongs to the key bladder cancer genes as recently defined in an international consensus paper [21]. Belonging to the EGFR-related receptor tyrosine kinase family, it is a key driver and well-established drug target in breast and gastric cancer. In our previous work, we showed that *ERBB2* mRNA expression is superior to the WHO grading of 1973 when dissecting the remaining risk in pT1 NMIBC exhibiting centrally confirmed grade 3 [22], with high *ERBB2* mRNA levels indicating inferior outcome (90% vs. 50% 5-year PFS, *p* < 0.0001). Higher levels are also associated with worse outcome in MIBC not being treated by adjuvant or neoadjuvant chemotherapy [23], with *ERBB2*-positive tumors above median mRNA expression having worse prognosis (20% vs. 60% 4-year DSS, *p* = 0.009). *ERBB2* mRNA is associated with luminal subtypes of bladder cancer [17,18,19].

However, the prognostic role of these markers in MIBC patients receiving adjuvant chemotherapy is unknown. The aim of the present study was to evaluate the prognostic role of the above-mentioned fundamental bladder cancer markers in the adjuvant situation and to test their clinical usefulness when assessed by IHC or PCR in context with clinical parameters to provide real-world evidence for the respective tumor biological motifs. The herein presented work served as a pilot study for validation of the above-mentioned biomarker assessment and moreover allowing the formulation of a working hypothesis for subsequent prospective non-interventional validation studies in the future. These studies are currently being planned and ultimately may lead to prospective interventional study designs.

## 2. Results

### 2.1. Patient Population

Clinical characteristics are presented in Table 1. The total study cohort consisted of 50 MIBC tumor patients diagnosed from 1996 to 2006 at a single institution. Median age was 65 years, with 76% male patients and 34% female patients; 50% of patients had ECOG status 0, while 34% and 16% were ECOG1 and ECOG2, respectively. Forty-two percent of patients were N0 at initial diagnosis, while 14% were N1 and 44% were N2. Median follow-up was 60.5 months with 54% of patients suffering from disease-specific deaths. Similar clinical characteristics were found in the analysis cohorts as defined in the cohort diagram (Table 1).

### 2.2. Distribution of Assessed Protein Markers across the Study Cohort

All investigated experimental markers could be determined by IHC or PCR in the same tissue microarray (TMA) samples of urinary bladder cancer transurethral resection of bladder (TURB) biopsies. 

### 2.3. Distribution of Assessed mRNA Markers across the Study Cohort

As depicted in the remark diagram (Figure 1), TURB biopsies from 39 patients could be analyzed, while IHC data were available from 28 TURB biopsies.

Data distribution of immunohistochemical staining of CD68, CLEVER-1, MAC387, and survivin by digital image analysis, visual inspection, or semi-quantitative assessment of cytoplasmic versus nuclear staining indicated a substantial infiltration of macrophages into the TURB biopsies of tumor specimens, while visual inspection reached higher sensitivity than image analysis. The numbers of CD68+ macrophages and CLEVER-1 positive macrophages and vessels were scored from three hotspots (areas with the most macrophages by eye) intratumorally and peritumorally with a 0.0625 mm2 grid using 40× magnification when scoring macrophages and 20× when scoring lymphatic/blood vessels. The scoring was performed independently by two observers blinded to the clinical information. Cases with an inadequate quality of immunohistochemical staining or tumor morphology were excluded from further statistical analyses. Survivin protein expression could be observed in almost all TURB biopsies with varying extent, while the nuclear staining of survivin could be detected in only 60% of cases (Figure 2a).

RNA expression of the candidate genes *KRT5, KRT20, ERBB2, BIRC5*, and *CD68* could also be detected to a varying extent. While *ERBB2* mRNA levels could be determined in almost all cases (38 of 39 samples), *KRT5* and *KRT20* mRNA were detected in fewer biopsies (31 of 39 and 24 of 39 samples, respectively). Similarly, *BIRC5* and *CD68* were detected in subsets of the TURB tissue dots (24 of 39 samples, each). A comparison of NMIBC and MIBC was possible for only four patients. Marker gene expression was comparable. However, with regard to *KRT20* expression, one MIBC did exhibit a significantly increased expression of the luminal marker *KRT20*.

### 2.4. Correlation of Protein and mRNA Markers on Basis of Molecular Subtyping and Clinical Variables

A comparison of survivin protein expression in cytoplasm versus nucleus compared to its respective mRNA level revealed that higher mRNA is positively associated to nuclear expression (Spearman rho 0.2949) and negatively associated with cytoplasmic stain (Spearman rho −0.3026), while both associations did not yet reach statistical significance due to small sample size (Figure 3).

As depicted in Figure 4a, Spearman correlation of the intergene RNA expression relations revealed a strong positive association between the two luminal cancer markers *KRT20* and *ERBB2* (Spearman rho 0.6811, *p* < 0.0001) and inverse relation between the luminal *KRT20* and basal *KRT5* marker (Spearman rho −0.3588, *p* = 0.0249) as expected. The negative association between *ERBB2* and *KRT5* was less prominent and not significant (Spearman rho −0.1473, *p* = 0.3709), indicating that several basal-like tumors harbor elevated *ERBB2* expression to some extent. Of note, the proliferation/apoptosis marker *BIRC5* was positively associated with *CD68* mRNA levels (Spearman rho 0.3484, *p* = 0.0156).

In line with this, *BIRC5* mRNA was also positively associated with CD68 levels determined by IHC (Spearman rho 0.4008, *p* = 0.0346; Figure 4b). Interestingly, infiltration by macrophages as determined by IHC of CD68 and MAC387 tended to be negatively associated with luminal tumors as determined by *KRT20* (Spearman rho −0.2720 and −0.1458) and positively with basal tumors as determined by *KRT5* (Spearman rho 0.1967 and 0.3489).

Pearson correlation of *KRT5, KRT20, ERBB2, BIRC5*, and *CD68* mRNA levels with clinical variables such as performance status (PS), age, sex, body mass index (BMI), presence of carcinoma in situ (Cis), tumor stage (T-prim), and WHO Grade 1973 (G-prim) levels in the larger PCR cohort (Figure 5a) revealed that luminal tumors determined by *KRT20* mRNA were negatively associated with the presence of Cis and positively associated with higher age and male gender. In contrast, basal tumors determined by *KRT5* were negatively associated with BMI. Interestingly, macrophage infiltration was positively associated with age and Cis, while being negatively associated with grade. *BIRC5* mRNA was comparably associated with Cis. Similar associations were obtained by doing Spearman correlations (Table 2).

### 2.5. Disease-Specific Survival Analysis by Survivin and Macrophage Infiltration in Subtypes

Kaplan–Meier analysis revealed that high levels of survivin protein above the median expression (>25% positive nuclei) in the IHC cohort (*n* = 28) identified patients with improved disease-specific survival (DSS, 60% vs. 10% 5-year DSS, *p* = 0.001; Figure 5a) and PFS (60% vs. 10% 5-year PFS, *p* = 0.0138; Appendix A).

Similarly, in the enlarged PCR cohort (n = 39), high levels of *BIRC5* mRNA (DCT >33.9) identified patients with better outcome (60% vs. 30% 5-year DSS, *p* = 0.0507; Figure 5b) and progression-free survival (75% vs. 10% 5-year PFS, *p* = 0.0042; Appendix A).

Combining survivin expression and *KRT20* mRNA for outcome prediction revealed that *KRT20*-positive tumors as well as *BIRC5*-negative tumors had worse outcomes compared to survivin-positive tumors both on the protein level (20% and 10% vs. 75% 5-year DSS, *p* = 0.0005; Figure 6a) and mRNA level (30% each vs. 75% 5-year DSS, *p* = 0.0358; Figure 6b). Similarly, the combination of *KRT20* mRNA with *BIRC5* mRNA or survivin protein stain was significant for PFS (*p* = 0.0181 Appendix A and *p* = 0.0209 Appendix A).

Multivariate cox proportional hazard analysis of DSS revealed that the combination of *BIRC5* and *KRT20* mRNA to predict outcome was an independent prognostic factor, when age, sex, BMI, tumor stage, grade, and node status were included in the analysis (*p* = 0.0167, Table 3). BMI and node status were also independent prognostic factors in multivariate cox regression. Similarly, multivariate cox proportional hazard analysis revealed that the combination of *BIRC5* and *KRT20* mRNA tended to be an independent prognostic factor for PFS (*p* = 0.0816; Table 4).

Combining survivin protein with the quantitation of macrophage infiltration based on protein or mRNA level revealed that the presence of macrophages in MIBC treated with adjuvant chemotherapy had an adverse effect on DSS. Tumors with high levels of nuclear survivin protein levels but low *CD68* mRNA levels had the best survival (70% vs. 40% vs. 10% 5-year DSS, *p* = 0.0083; Figure 7a). Similar results were found for PFS (*p* = 0.0169, Appendix A).

In line with this, tumors with high levels of nuclear survivin protein levels but low MAC387 protein levels had the best survival (100% vs. 30% vs. 10% 5-year DSS, *p* = 0.0011; Figure 7b). Similar results were found for PFS (*p* = 0.0259, Appendix A). Additionally, Appendix A shows DSS of bladder cancer patients treated with adjuvant chemotherapy based on survivin nuclear protein stain and CLEVER-1 protein in the PCR and IHC cohort. Appendix A shows PFS of bladder cancer patients treated with adjuvant chemotherapy based on survivin nuclear protein stain and CLEVER-1 protein in the PCR and IHC cohort.

Importantly and in contrast to previous publications, in pT1 NMIBC [22] and MIBC [23] not treated with chemotherapy, the overexpression of *ERBB2* was not related to adverse outcome (data not shown). However, within *ERBB2*-positive tumors (median mRNA expression), the presence of macrophages as determined by RT-qPCR of *CD68* had an adverse effect on the DSS of adjuvant-treated MIBC patients (70% vs. 20% 5-year DSS, *p* = 0.0280; Figure 8). Similarly, the combination of *ERBB2* and *CD68* tended to predict PFS (*p* = 0.0537, Appendix A).

## 3. Discussion

High levels of survivin have been associated with poor prognosis in bladder cancer [24]. Survivin has also been described to be a predictor of cisplatin-resistance in gastric cancer, as well as in different cell lines [25,26]. A higher proliferative activity determined by *BIRC5* mRNA expression has been associated with worse outcome in NMIBC [11]. In line with this, a higher WHO 1973 grade was associated with *MKI67* and *ERBB2* mRNA levels [20]. Similarly, *FOXM1* mRNA expression was associated with a higher grade and stage as well as a 6 to 8-fold higher risk of progression in multivariable analysis (*p* < 0.03) of the UROMOL study (n = 488), which could be validated in independent NMIBC cohorts (n = 277) in silico [27]. Further analysis revealed that proliferation as determined by *FOXM1* mRNA expression was predictive for chemotherapy benefit in T1 NMIBC (n = 296) with patients having low *FOXM1* expression having better outcomes, irrespective of instillation therapy, while patients with high *FOXM1* expression benefitted from intravesical chemotherapy with mitomycin C [28]. In addition, meta-analysis revealed survivin protein and RNA to be associated with adverse outcome in NMIBC [29]. However, the predictive or prognostic role of proliferation and particularly of survivin is less clear for MIBC, particularly upon chemotherapeutic intervention targeting proliferative tissues. We showed that a high expression of survivin both on protein and RNA level was associated with good outcome in MIBC patients treated with adjuvant chemotherapy. It has to be noted that the triple chemotherapeutic regimen investigated within this study including taxol in addition to platinum-based chemotherapy is no standard regimen, which has to be taken into account when interpreting the results. However, it is reasonable that highly proliferative tissues do exhibit better response to chemotherapeutic regimen. Moreover, it has to be assumed that adding taxol to the standard chemotherapeutic regimen does not diminish the non-response of tumor tissues with low proliferative activity reflected by low survivin expression. This indicates that survivin might be a good predictive marker for chemotherapy benefit, which should be further investigated in randomized clinical trials. In contrast, low levels of nuclear staining of survivin were associated with the DSS of only 10% of patients after 5 years (*p* = 0.001), which indicates that tumors with low proliferation and apoptotic activity as indicated by survivin expression do require alternative treatment approaches. 

In contrast, Als et al. identified survivin as a molecular marker for survival in locally advanced and/or metastatic bladder cancer following cisplatin-based chemotherapy [30]. In their study, multivariate analysis revealed that survivin expression was an independent marker for poor outcome, together with the presence of visceral metastases. In the group of patients without visceral metastases, both markers showed significant discriminating power as supplemental risk factors (*p* < 0.0001). Protein expression assessed by IHC was strongly correlated to response to chemotherapy. Another study on survivin was published by Pollard et al. [31]. This group evaluated an approach that combines genomic, proteomic, and therapeutic outcome datasets to identify novel putative urinary biomarkers of clinical outcome after neoadjuvant application of methotrexate, vinblastine, adriamycin, and cisplatin (MVAC). Using disease-free survival as a marker for clinical outcome, this group evaluated the ability of GGH, emmprin, survivin, and DBI expression in tumor tissue to stratify 27 patients treated with neoadjuvant MVAC. Interestingly, DBI (*p* = 0.046) but not GGH (*p* = 0.190), emmprin (*p* = 0.066), or survivin (*p* = 0.393) successfully stratified patients [31]. Our study revealed an inverse relation of survivin protein in cytoplasmic versus nuclear localization particularly when compared to its mRNA levels. This indicates the need of careful subcellular quantitation and may in part explain conflicting study results with regard to the prognostic and or predictive value of survivin expression, as discussed above.

Importantly, in our study, the proliferative subset of MIBC patients having better survival (i.e., 60% DSS after 5 years) could be further dissected by macrophage infiltration. Tervahartiala et al. [14] found that MAC387+ cells as well as CLEVER-1+ macrophages and vessels are associated with the response after neoadjuvant chemotherapy in bladder cancer patients. High MAC387+ tumor cell density was associated with disease progression after neoadjuvant chemotherapy, whereas the majority of patients with a lower amount of MAC387+ tumor cells exhibited a complete response. Patients with high amounts of CLEVER-1+ macrophages were associated with a poorer response to neoadjuvant chemotherapy, while higher amounts of CLEVER-1+ vessels were associated with a more favorable response [14]. The results of Tervahartiala et al. [14] verified also their previous studies where they could demonstrate that CD68 and MAC387 are associated with poorer survival in bladder cancer patients, whereas CLEVER-1-positive vessels act more as a protective marker [32].

In our study, we could validate that the presence of macrophages as determined by immunohistochemistry of CD68, CLEVER-1, and MAC387 or PCR of *CD68* was associated with worse disease-specific survival, particularly in tumors of high proliferative activity or elevated *ERBB2* mRNA expression.

Macrophages are challenging to investigate by immunohistochemistry due to their nature to cluster. This may lead to variations in results, especially when using TMAs and would require sufficient tissue sampling in routine clinical practice. TMAs are an efficient method in immunohistochemistry, but the results should be interpreted with care when studying clustering particles, e.g., macrophages. RNA quantitation may offer advantages by a more objective and standardized assessment of macrophage infiltration and the opportunity to embed the results in the context of immune infiltrates of diverse sets of T-cells with specified functions such as natural killer cells, helper cells, and regulatory T-cells.

The potential limitations of our study relate to its retrospective design and the impact of factors such as age and comorbidity on the indication of cystectomy and, consequently, on cancer-specific mortality in the elderly patients. The number of patients was limited, but the study included consecutive bladder cancer patients, who received adjuvant chemotherapy after radical cystectomy. Since retrospective designs do not guarantee causality, further prospective studies and the use of an independent series are warranted to prove the prognostic and predictive value of the analyzed marker combinations to robustly stratify the clinical outcome in real-world assessments.

## 4. Materials and Methods

### 4.1. Patients

#### 4.1.1. Patient Population

From August 1996 to June 2006, a total of 50 patients diagnosed with bladder cancer were included in the trial. Together, 38 male patients and 12 female patients (average age 65 years, range 49–80 years) were included. Pathohistological T-category and grade for the primary tumors are as follows. The study included for the primary tumors pTaG2 (n = 1), pT1G2 (n = 9), pT1G3 (n = 7), pT2G1 (n = 1), pT2G2 (n = 10), and pT2G3 (n = 22) obtained by transurethral resection under institutional review board-approved protocols. Three patients showed carcinoma in situ (6%). All non-muscle invasive urothelial carcinomas included in the study progressed to muscle-invasive tumors under the follow-up. All patients were treated with radical surgery before chemotherapy. Patient characteristics, including lymph node status before chemotherapy as well as ECOG performance status at the point of starting chemotherapy, are summarized in Table 1. The study population had its origin in one single institution. The analysis of the different markers has been performed at different study sites.

#### 4.1.2. Eligibility

Eligible patients for this trial were required to have either metastatic or locally advanced histologically confirmed transitional cell carcinoma of the urothelial tract. Patients who had received a previous systemic chemotherapy regimen were excluded. Previous radiation therapy was also an exclusion criterion.

Additional eligibility requirements included the following: an ECOG performance status of 0 to 2, a leukocyte count ≥3000/µL, a platelet count ≥100,000/µL, serum bilirubin <1.5 mg/dL, serum creatinine ≤2.5 mg/dL, and age >18 years. Patients with other active malignancies or any other serious or active medical conditions were excluded. Pregnant or lactating females were ineligible. The study protocol was approved by the Research Ethical Board of the Landesärztekammer Brandenburg (AS 25(bB)/2017; AS 147(bB)/2013) for the German part of the study. For the Finnish part, there was an ethical approval from the Hospital District of Southwestern Finland. All methods in this study were carried out in accordance with relevant guidelines and regulations. The study was conducted in compliance with the current revision of the Declaration of Helsinki, guiding physicians and medical research involving human subjects. All patients were required to provide written informed consent prior to the study enrolment. The study did not affect the patients or their further treatment of follow-up in any way. All the sample collections were done on already existing tissue specimens received during the diagnosis and treatment of these patients. 

### 4.2. Pretreatment Evaluation

Prior to enrollment in this trial, all patients were required to have a complete history, physical examination, complete blood counts, chemistry profile, and urine analysis. In addition, patients underwent computed tomography scans of the chest, abdomen, and pelvis with appropriate tumor measurements.

### 4.3. Assessment of Treatment Efficacy

All fifty patients received treatment with the following regimen: gemcitabine at a dose of 1000 mg/m^2^ as a 30 min intravenous infusion followed by paclitaxel at a dose of 80 mg/m^2^ as a 1 h intravenous infusion on days 1 and 8. On day 2, cisplatin at a dose of 50 mg/m^2^ was administered as an intravenous infusion and hydration with 2000 mL NaCl 0.9%. The regimen was repeated every 21 days. Patients received standard paclitaxel premedication and antiemetic prophylaxis. Patients were evaluated for response to treatment after the completion of 4 courses (12 weeks). Reevaluation included a repeat of all previously abnormal radiologic studies with a repeat of objective tumor measurement. Patients who achieved an objective response (complete or partial) or stable disease after the completion of four courses of therapy continued treatment with this regimen. Treatment was continued for a total of six courses. None of the patients received neoadjuvant therapy before cystectomy.

Thirty-four patients who completed 6 courses and remained in remission were followed with further treatment of a single dose of gemcitabine at a dose of 1000 mg/m^2^ as a 30 min intravenous infusion repeating every 28 days. This following treatment was continued for at least two years.

Two patients received the second-line chemotherapy (methotrexate, epirubicin and cisplatin chemotherapy (MEC)) because of rapid progression after a three-drug regimen with gemcitabine, paclitaxel, and cisplatin or during gemcitabine monotherapy.

### 4.4. Dose Modifications

All patients received full doses of all 3 agents on day 1 of the first course of treatment. Subsequent doses were based on hematologic and non-hematologic toxicity observed. Dose modifications for myelosuppression were determined by the blood counts measured on the day of scheduled treatment. Nadir blood counts were not used as a basis for dose reduction.

On day 1 of each course, full doses of all drugs were administered if the leukocyte count was ≥3000/µL and the platelet count was >100,000/µL. If the leukocyte count was <3000/µL or the platelet count was <100,000/µL, treatment was delayed for one or two days.

All patients with an ECOG performance status of 2 or with renal insufficiency in the stage of compensated retention received reduced doses of 50% to 70%. In case of good tolerance of the therapy, we applicated higher doses for following cycles.

### 4.5. Criteria for Follow-Up

The follow-up consisted of clinical examination, ultrasound of abdomen, and computed tomography scans of the chest, abdomen, and pelvis with appropriate tumor measurements every 6 months. Progression was defined as new metastatic disease or local progress during follow-up. Chemotherapy response was defined as absence of recurrence, progression, or death from the disease during follow-up. Responses were defined using the Response Evaluation criteria in Solid Tumors (RECIST). A complete response (CR) required the total disappearance of all clinically and radiographically detected tumors for at least 4 weeks. Patients had partial response (PR) if treatment produced a reduction of at least 30% in the sum of the longest diameter, with no evidence of new disease. No change (NC) was defined as patients who showed no visible reduction or even progress less than 20%. Patients who had the appearance of any new lesions or who had an increase of at least 20% in the size of any existing lesions had progressive disease (PD).

### 4.6. Clinical Follow-Up and Treatment Efficacy

Among the 50 cases analyzed, 26 progressed (52%), and 34 patients died (68%). Tumor-related death is 27 in total (54%). Twenty-one patients (42%) achieved complete response, three patients achieved partial response (6%), and for twenty-five patients (50%), no change was documented (Table 1).

The median time interval from diagnosis at the point of the first transurethral resection of the primary tumor and the date of death (or last follow-up) of all patients was 36.5 months (range: 8.0–221.0). The median time interval between the point of radical operation of the muscle-invasive tumor and the date of death (or last follow-up) of all patients was 49.0 months (range: 7.0–175.0). The median time interval between the point of chemotherapy and the date of death (or last follow-up) of all patients was 23.0 months (range: 3.0–171.0). The median time between primary resection and muscle-invasive tumor at the point of radical operation of all patients was 8.0 months (range: 6.0–58.0). The median time to progress between radical operation and the progressive disease of all patients was 13 months (range: 6.0–32.0).

### 4.7. Procedure

For each case, the most representative formalin-fixed, paraffin-embedded tissue block was selected for analysis. Sections (5 μm thickness) were deparaffinized with xylene and rehydrated with a graded alcohol series.

### 4.8. Immunostaining for CD68, MAC387, and CLEVER-1

The primary antibodies used were mouse monoclonal IgG1 antiCD68 (KP1) (concentration 1:5; ab845, Abcam, U.K.) and mouse monoclonal IgG1 antiMAC387 (concentration 1:500; ab22506, Abcam, U.K.), which detects the myelomonocytic L1 molecule calprotectin. CLEVER-1 (common lymphatic endothelial and vascular endothelial receptor-1, also known as STAB1 and FEEL-1) positive type 2 macrophages and vessels were detected with the rat IgG 2-7 antibody (concentration 1:5) [33,34]. The antibodies 3G6 (mouse IgG1 antibody against chicken T cells) [35] and MEL-14 [rat IgG2a antibody against mouse L-selectin (CD62L)] (Exbio, Czech Republic) were used as negative controls. The primary immunoreaction was performed with using the mouse/rat Vectastain Elite ABC Kit (Vector Laboratories). Sections for CD68 and MAC387 staining were heat pre-treated in citric acid (0.01 M, pH 6.0) in a 97 °C water bath for 20 min. Antigen retrieval for CLEVER-1-stained sections was performed with proteinase K (Dako, Glostrup, Denmark) (10 min at 37 °C), and the slides were washed three times with PBS after the pre-treatment. Endogenous peroxidase was blocked with 0.1% H_2_O_2_ for 30 min. Non-specific sites were blocked with horse (CD68 and MAC387) or rabbit (CLEVER-1) normal serum at room temperature for 20 min. Sections were incubated with primary antibodies overnight at 4 °C and then treated with biotinylated secondary antibody solution according to the manufacturer’s instructions. After washing with PBS, Vectastain Elite ABC Reagent was added (30 min at room temperature), the slides were washed, and immunoreactions were detected using 3,3′-diaminobenzidine as a substrate. Slides were counterstained with hematoxylin, dehydrated, re-fixed in xylene, mounted with distyrene plasticizer xylene (DPX). The whole tumor and surrounding peritumoral area were screened by light microscopy. A detailed description of that scoring process has already been published by Boström et al. [32]. These experiments have been performed at University Hospital Turku (Finland).

### 4.9. Immunostaining for Survivin

Survivin antibody was provided by the Department of Molecular Medicine at the Institute for Prevention and Occupational Medicine of the German Social Accident Insurance in Bochum, Germany. A detailed description of recombinant survivin and antibody production as well as the analytical specificity of the survivin antibody can be found at Gleichenhagen et al. [36]. We used this method for survivin in a regular immunohistochemistry procedure. In this study, we are one of the first centers who evaluated this survivin antibody by immunohistochemistry. Reproducibility of the new survivin antibody as a component of an ELISA was investigated and published by Gleichenhagen et al. [36]. The experiments for immunostaining for survivin have been performed and evaluated at University Hospital Charité Berlin (Germany). Examples of immunohistochemical stainings are visible in Appendix A.

### 4.10. Isolation of Tumor RNA

For RNA extraction from FFPE tissue, tissue dots (1.5 mm diameter, 5 µm cuts) from tissue microarray material were picked by stamp technology and further processed according to a commercially available bead-based extraction method (XTRACT kit; STRATIFYER Molecular Pathology GmbH, Cologne, Germany). RNA was eluted with 100 μL elution buffer, and then, RNA eluates were stored at −80 °C until use. 

### 4.11. Gene Expression by RT-qPCR 

The mRNA expression levels of *KRT5, KRT20, ERBB2, BIRC5* and *CD68* as well as one reference gene (REF), namely *CALM2*, were determined by RT-qPCR, which involves the reverse transcription of RNA and subsequent amplification of cDNA executed successively as a 1-step reaction using inventoried validated TaqMan Gene Expression Assays (MP002, MP015, MP452, MP089, MP120 and MP501, STRATIFYER Molecular Pathology GmbH, Köln, Germany). The robustness and usefulness of *CALM2* as a housekeeping gene for diverse candidate genes as well as comparability to diverse IHC assessments such as CK20/KRT20, MKI67/Ki67, and PDL1, when used as a single reference gene, have been demonstrated in several of our own publications [20,37,38] and resulted in the introduction of *CALM2* as a housekeeping gene in CE-certified IVD products such as Endopredict [39] and MammaTyper [40]. Each patient sample or control was analyzed with each assay mix in triplicate. The experiments were run on a Siemens Versant (Siemens, Germany) according to the following protocol: 5 min at 50 °C, 20 Sec at 95 °C, followed by 40 cycles of 15 Sec at 95 °C and 60 Sec at 60 °C. Forty amplification cycles were applied, and the cycle quantification threshold (Cq) values of three markers and one reference gene for each sample (S) were estimated as the median of the triplicate measurements. The final values were generated using ΔCT from the total number of cycles (40-DCT) to ensure that the normalized gene expression obtained by the test was proportional to the corresponding mRNA expression levels. This part of the work has been measured and analyzed at STRATIFYER Molecular Pathology, Cologne (Germany). Examples of immunostaining for survivin, CD68, CLEVER-1, and MAC387 are shown in Appendix A.

### 4.12. Statistical Analysis

The Kaplan–Meier method, log-rank testing, and Cox proportional hazards regression models were used to analyze the associations between IHC and outcome. Partitioning tests were used to identify appropriate cut-off values for dichotomization of the continuous variables for Kaplan–Meier analysis. In the Cox proportional hazards regression models, the markers were evaluated as continuous variables. Outcome measures included DSS and OS. The survival time was calculated from the date of surgery to the date of the last follow-up or death. Any death due to bladder cancer (BC) or with metastatic BC was defined as cancer-specific mortality. All statistical tests were two-sided, and p-values <0.05 were considered statistically significant. All tests and calculations were performed using the software R, version 3.1.2 (R Development Core Team 2014) or JMP 9.0.0 (SAS Institute Inc, 100 SAS Campus Drive Cary, NC 27513-2414, USA).

## 5. Conclusions

Markers that are validated to predict poor prognosis in NMIBC and MIBC not being treated with chemotherapy, such as survivin and potentially *ERBB2*, exhibit inverse outcome relation upon adjuvant chemotherapeutic treatment, indicating their potential as being predictive for chemotherapy benefit. In addition, macrophage infiltration seems to have a key role in high-risk tumors that could be attributed to its potential of modulating the activity of infiltrating T-cells particularly under circumstances of the chemotherapeutic destruction of tumor cells and subsequent antigen presentation. The findings of the study are limited by the small size of the stud group and its retrospective character, but based on its results, we could demonstrate that further prospective studies with a higher number of patients might be worth pursuing. The combination of immunohistochemical and robust molecular methods being applicable in clinical routine situation harbors the promise of predicting the outcome of patients and serving as valuable pathological tools to better select patients for specific therapeutic interventions.

## Figures and Tables

**Figure 1 ijms-21-07420-f001:**
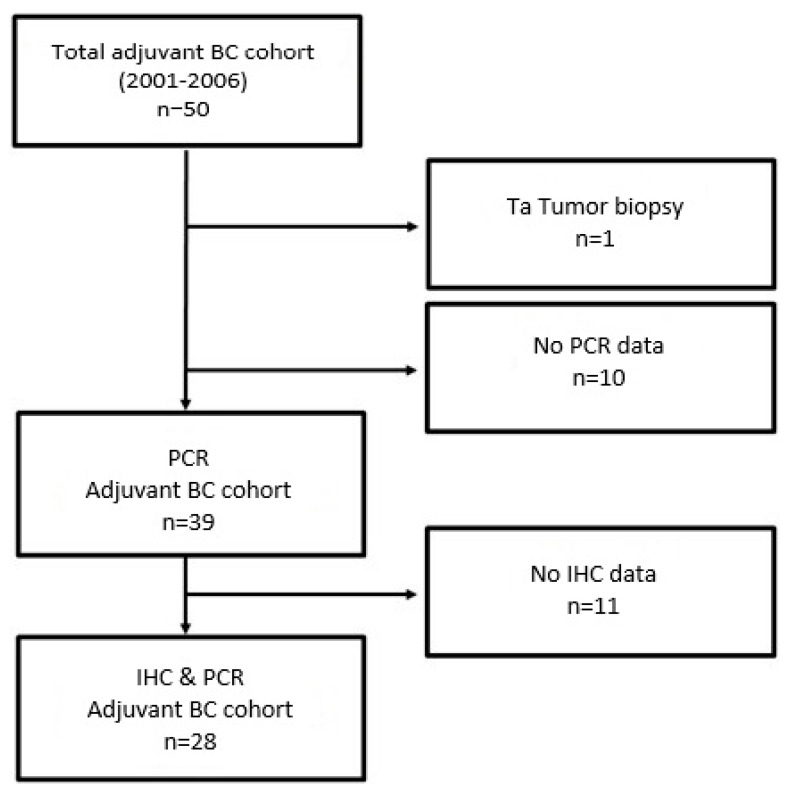
Remark diagram.

**Figure 2 ijms-21-07420-f002:**
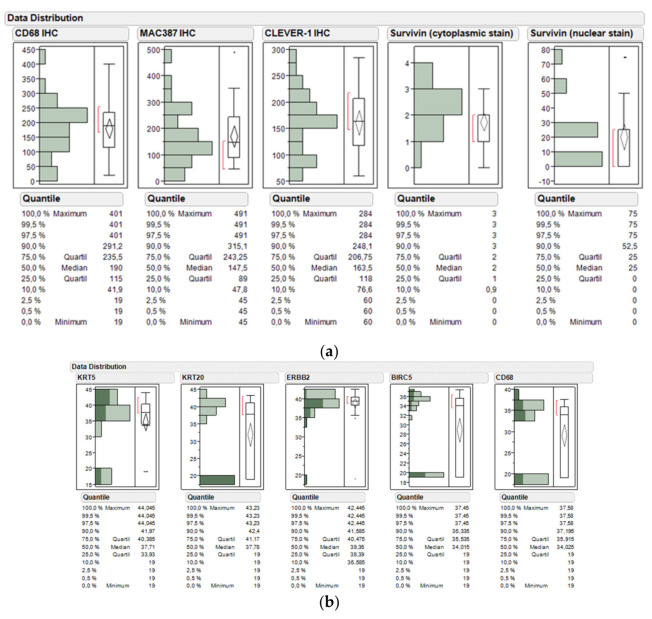
(**a**) Data distribution of immunohistochemical staining of CD68, MAC387, and common lymphatic endothelial and vascular endothelial receptor-1 (CLEVER-1) by visual analysis and survivin by semi-quantitative assessment of cytoplasmic versus nuclear stain; (**b**) Data distribution and box and whisker plot of *KRT5*, *KRT20*, *ERBB2*, *BIRC5*, and *CD68* mRNA levels in the bladder cancer study cohort treated by adjuvant chemotherapy (*n* = 39). Normalized gene expression (40-DCT method) as well as quantile values are depicted in the y-axis. DCT: Delta Cycle Threshold.

**Figure 3 ijms-21-07420-f003:**
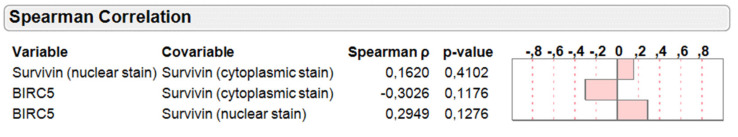
Spearman correlation of IHC staining and semi-quantitative assessment of survivin protein located in cytoplasmic versus nuclear localization with quantitative *BIRC5* (survivin) mRNA levels in the combined PCR and IHC cohort (*n* = 28). Graphical display of Spearman rho values and respective *p*-values are depicted.

**Figure 4 ijms-21-07420-f004:**
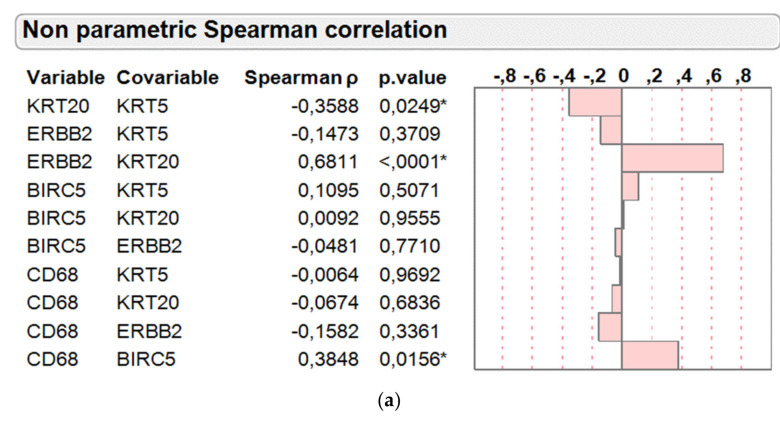
(**a**) Correlation of normalized *KRT5*, *KRT20*, *ERBB2*, *BIRC5*, and *CD68* mRNA levels in the PCR cohort (*n* = 39) of bladder cancer patients treated with adjuvant chemotherapy. Graphical display of Spearman rho values and respective p-values are depicted. * indicates statistically significant results; (**b**) Correlation of *KRT5*, *KRT20*, *ERBB2*, and *BIRC5* mRNA levels with protein levels of CD68, MAC387, and CLEVER-1 determined by IHC in the combined PCR and IHC cohort (*n* = 28) of bladder cancer patients treated with adjuvant chemotherapy. Graphical display of Spearman rho values and respective p-values are depicted. * indicates statistically significant results.

**Figure 5 ijms-21-07420-f005:**
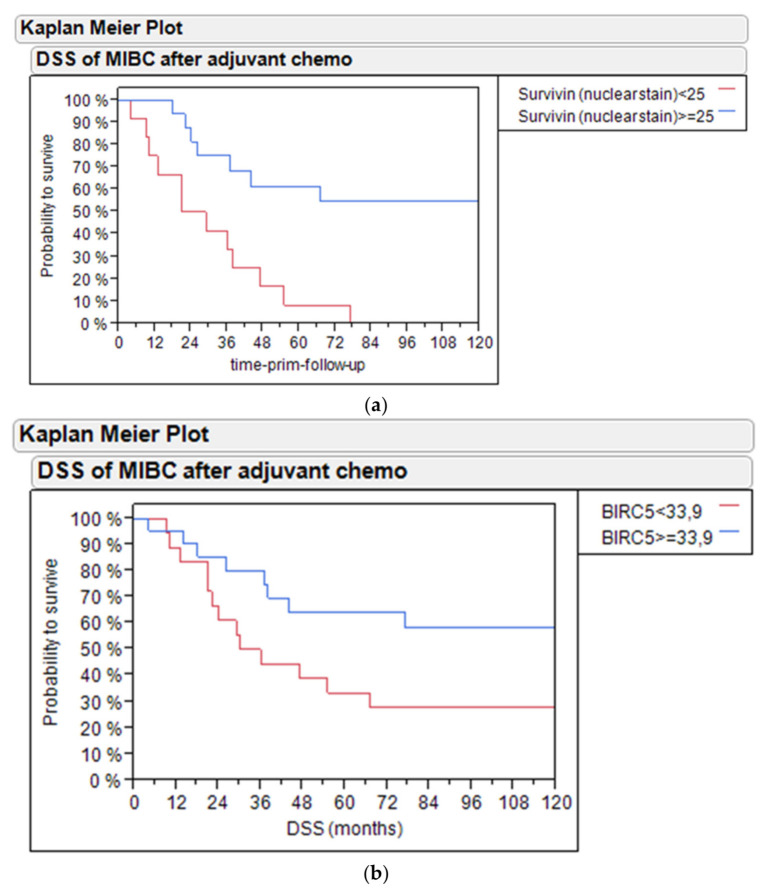
(**a**) Disease-specific survival (DSS) of bladder cancer patients treated with adjuvant chemotherapy based on survivin nuclear stain in the PCR and IHC cohort. (**b**) DSS of bladder cancer patients treated with adjuvant chemotherapy based on *BIRC5* mRNA expression in the PCR cohort.

**Figure 6 ijms-21-07420-f006:**
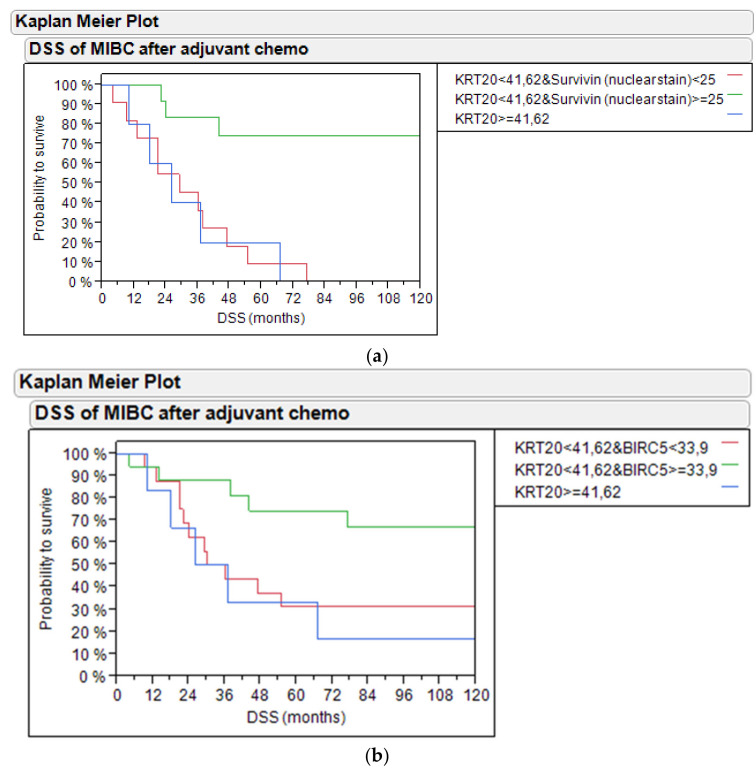
(**a**) DSS of bladder cancer patients treated with adjuvant chemotherapy based on *KRT20* mRNA and survivin nuclear protein stain in the PCR and IHC cohort; (**b**) DSS of bladder cancer patients treated with adjuvant chemotherapy based on *KRT20* and *BIRC5* mRNA expression in the PCR cohort.

**Figure 7 ijms-21-07420-f007:**
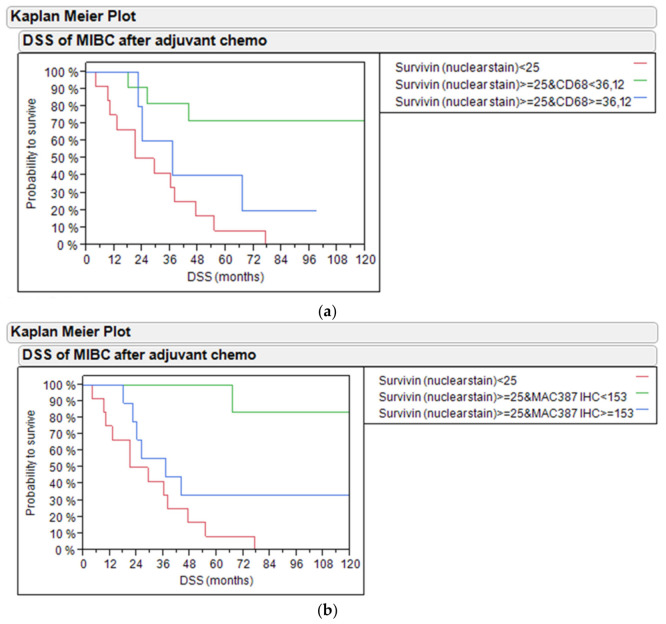
(**a**) DSS of bladder cancer patients treated with adjuvant chemotherapy based on survivin nuclear protein staining and *CD68* mRNA in the PCR and IHC cohort. (**b**) DSS of bladder cancer patients treated with adjuvant chemotherapy based on survivin nuclear protein staining and MAC387 protein in the PCR and IHC cohort.

**Figure 8 ijms-21-07420-f008:**
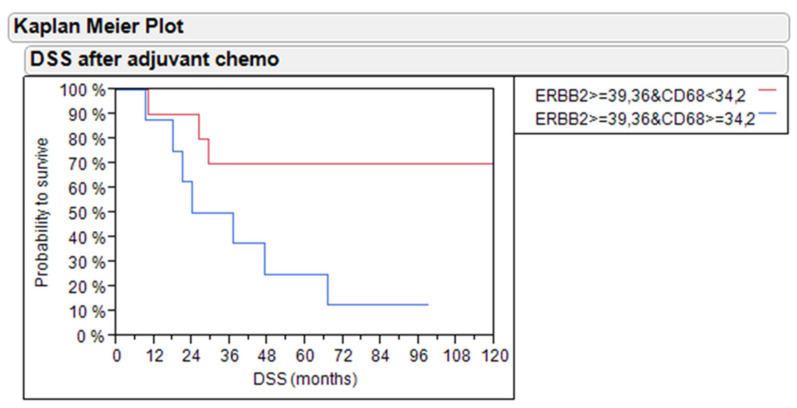
DSS of bladder cancer patients treated with adjuvant chemotherapy based on *ERBB2*-positive tumors in relation to *CD68* mRNA levels in the PCR and IHC cohort.

**Table 1 ijms-21-07420-t001:** Clinical characteristics of patients in the total cohort (*n* = 50), and the PCR (*n* = 39) and combined IHC and PCR subcohorts (*n* = 28). IHC: immunohistochemistry.

Cohort	Total Cohort	PCR Cohort	IHC & PCR Cohort
Size (*n*)	50	39	28
**Age (years)**			
Average	65	67	68.5
Range	49-80	48–80	48–80
**Gender**			
Male	38 (76%)	27 (69%)	18 (64%)
Female	12 (24%)	12 (31%)	10 (36%)
**ECOG Performance Status**			
0	25 (50%)	19 (49%)	11 (39%)
1	17 (34%)	13 (33%)	11 (39%)
2	8 (16%)	7 (18%)	6 (21%)
**Lymph Node Metastases before Chemotherapy**			
N0	21 (42%)	16 (41%)	10 (36%)
N1	7 (14%)	4 (10%)	2 (7%)
N2	22 (44%)	(19 (49%)	16 (57%)
**Clinical outcome after Chemotherapy**			
Progression	27 (54%)	21 (54%)	18 (64%)
Overall death	36 (72%)	29 (74%)	23 (82%)
Disease specific death	27 (54%)	21 (54%)	19 (68%)
Overall survival	14 (28%)	10 (26%)	5 (18%)
**Response to Chemotherapy**			
Complete response	20 (40%)	15 (38%)	8 (29%)
Partial response	3 (6%)	2 (5%)	1 (4%)
No change	25 (50%)	20 (51%)	18 (64%)

**Table 2 ijms-21-07420-t002:** Pearson correlation of *KRT5*, *KRT20*, *ERBB2*, *BIRC5*, and *CD68* mRNA levels with performance status (PS), age, sex, body mass index (BMI), presence of carcinoma in situ (Cis), tumor size (T-prim), and WHO Grade 1973 (G-prim) levels in the PCR cohort of bladder cancer patients treated with adjuvant chemotherapy. Blue values indicate positive associations of significance, red values indicate negative associations of significance, and black values indicate insignificant trends.

	KRT5	KRT20	ERBB2	BIRC5	CD68	PS	Age	Sex	BMI	Cis	T-Prim	G-Prim
KRT5	1.0000	−0.1522	−0.0286	0.1052	0.0028	−0.0493	−0.0477	0.1678	−0.2302	−0.0040	−0.1413	0.0691
KRT20	−0.01522	1.0000	0.4266	0.0763	−0.1783	−0.0443	0.2165	0.2627	0.0498	−0.3599	−0.0544	0.1547
ERBB2	−0.0296	0.4266	1.0000	0.1019	0.0507	−0.2721	0.0280	0.3259	−0.0149	−0.0934	−0.1754	−0.0563
BIRC5	0.1052	0.0763	0.1019	1.0000	0.5390	−0.0579	0.2019	0.1273	−0.0553	0.2807	0.1831	0.0858
CD68	0.0028	−0.1783	0.0507	0.5390	1.0000	0.1646	0.3190	−0.0525	−0.1784	0.2361	0.1812	−0.2662
PS	−0.0493	−0.0443	−0.2721	−0.0578	0.1646	1.0000	0.3352	−0.1978	0.0978	−0.1370	0.1196	−0.0217
Age	−0.0477	0.2185	0.0280	0.2019	0.3190	0.3352	1.0000	0.1101	−0.3689	0.1915	−0.1977	−0.0184
Sex	0.1678	0.2827	0.3259	0.1273	−0.0625	−0.1978	0.1101	1.0000	0.0990	−0.0160	−0.1514	−0.0358
BMI	−0.2302	0.0498	−0.0149	−0.0553	−0.1784	0.0978	−0.3689	0.0990	1.0000	−0.2538	0.2724	−0.0904
Cis	−0.0040	−0.3599	−0.0934	0.2007	0.2361	−0.1370	0.1915	−0.0160	−0.2538	1.0000	−0.1852	0.0154
T-prim	−0.1413	−0.0544	−0.1754	0.1831	0.1612	0.1196	−0.1977	−0.1514	0.2724	−0.1852	1.0000	0.2200
G-prim	0.0691	0.1547	−0.0563	0.0858	−0.2662	−0.0217	−0.0184	−0.0356	−0.0904	0.0154	0.2200	1.0000

**Table 3 ijms-21-07420-t003:** Cox regression analysis for DSS by *BIRC5* × *KRT20* mRNA expression and clinicopathological features in the PCR cohort of bladder cancer patients treated with adjuvant chemotherapy. Statistically significant values are highlighted in boldface.

Parameter	Hazard Ratio	95% CI	*p*-Value
Age	1.07	0.99–1.14	0.0518
Sex	0.89	0.27–2.98	0.8329
BMI	1.16	1.00–1.24	**0.0499**
Node status	1.95	1.15–3.53	**0.0127**
Stage	1.13	0.42–3.16	0.8038
Grade	0.86	0.33–2.29	0.7638
**KRT20 × BIRC5 Groups**			
KRT20 low & BIRC5 high vs. KRT20 low & BIRC5 low	0.22	0.06–0.75	**0.0144**
KRT20 low & BIRC5 high vs. KRT20 high	0.24	0.06–0.94	**0.0407**
KRT20 low & BIRC5 low vs. KRT20 high	1.09	0.28–4.39	0.8988

**Table 4 ijms-21-07420-t004:** Cox regression analysis for progression-free survival (PFS) by *BIRC5* × *KRT20* mRNA expression and clinicopathological features in the PCR cohort of bladder cancer patients treated with adjuvant chemotherapy. Statistically significant values are highlighted in boldface.

Parameter	Hazard Ratio	95% CI	*p*-Value
Age	1.08	0.94–1.25	0.2911
Sex	1.25	0.23–7.45	0.7979
BMI	1.19	0.97–1.47	0.0985
Node status	1.75	0.89–3.87	0.1045
Stage	0.65	0.19–2.11	0.4750
Grade	0.86	0.33–2.29	0.4989
**KRT20 × BIRC5 Groups**			
KRT20 low & BIRC5 high vs. KRT20 low & BIRC5 low	0.15	0.02–0.79	**0.0252**
KRT20 low & BIRC5 high vs. KRT20 high	0.26	0.02–1.93	0.1908
KRT20 low & BIRC5 low vs. KRT20 high	1.77	0.41–8.92	0.4489

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
