# Peer review of "Prognostic Role of Survivin and Macrophage Infiltration Quantified on Protein and mRNA Level in Molecular Subtypes Determined by RT-qPCR of KRT5, KRT20, and ERBB2 in Muscle-Invasive Bladder Cancer Treated by Adjuvant Chemotherapy"

_ijms, 2020, doi:10.3390/ijms21197420_

Round 1

Reviewer 1 Report

This is important work indicating new combinations of the markers for outcome prediction and selection an optimal therapies. The idea of combining protein and RNA markers from the same samples is attractive. However, the source of the samples is FFPE tissue. It would be very helpful to compare these profiles with biomarkers obtained from a liquid biopsy. 

There are no adequate controls in this study because of the experimental design (tissue samples) and as the authors mentioned, the total number of patients is low. The retrospective nature of the study is another limitation that can be partly overcome with liquid biopsy collection.

Connection to biomarkers to molecular subtypes is very important. However, macrophage infiltration markers may be skipped in the current study and used for another trial and combined latter. The statistical methods are adequate and convincing, but too many of them. So it would be easier for reader to see it in a more condensed form. In summary, this is an important and well-designed study. 

Author Response

This is important work indicating new combinations of the markers for outcome prediction and selection an optimal therapies. The idea of combining protein and RNA markers from the same samples is attractive. However, the source of the samples is FFPE tissue. It would be very helpful to compare these profiles with biomarkers obtained from a liquid biopsy.

Answer: Thanks a lot for that helpfully comment. Indeed, liquid biopsies are a really interesting field in that context. Particularly with regard to mutational aberrations being found in primary tumors and recapitulated for monitoring during  follow up. However, we have focussed on RNA analytics using RNA expression profiles of distinct subtyping marked and therapy targets from FFPE tissues of initial biopsies to further develop new candidates that aid in initial therapy planning.  These markers are rarely mutated, so that it is not possible to detect them by DNA analytics from liquid biopsies. Circulating tumor cells would be an additional opportunity but subtyping and target quantitation from circulating tumor cells seems to be even more demanding from our perspective and therefore hast yet not been in our research focus.

There are no adequate controls in this study because of the experimental design (tissue samples) and as the authors mentioned, the total number of patients is low. The retrospective nature of the study is another limitation that can be partly overcome with liquid biopsy collection.

Answer: We fully agree with the reviewer. In most cases it is essential to have healthy controls as reference. However, controls as reference are more important in analysis of mutations in selected genes. In that cases liquid biopsies could also be helpful. In future studies we could include such samples. It is also correct that the patient number is low. However, for the molecular assessments we have used validated assay systems, which have already been analyzed in large trials of non-muscle invasive and muscle invasive bladder cancer (e.g. Breyer et al Virchows Arch 2017, Breyer et al Oncotarget 2017, Kriegmaier et al Trans Oncol 2018, Eckstein et al Oncotarget 2018). Therefore by recapitulating  the intergene associations in this smaller cohort we could demonstrate the representativeness of this smaller cohort, which on the other hand has the advantage of very homogenous adjuvant treatment and long term follow up, which had not been investigated in the previous studies. This allowed us to develop hypothesis generating hypothesis and also prove first aspects, despite the small study cohort by Kaplan Meier statistics.

Connection to biomarkers to molecular subtypes is very important. However, macrophage infiltration markers may be skipped in the current study and used for another trial and combined latter. The statistical methods are adequate and convincing, but too many of them. So it would be easier for reader to see it in a more condensed form. In summary, this is an important and well-designed study.

Answer: The comment of the reviewer is right. There are many information and statistics in our manuscript. However, we tried to focus on the most important details. We decided to include also the really interesting findings in macrophage infiltrations as this contributes to a new understanding that not only T-cell infiltration and their control by check point mechanisms is relevant but also the antigen presenting macrophages exhibit  an important function particularly upon chemotherapeutic treatment and responses thereto.

Reviewer 2 Report

Authors investigated the role of proliferation survivin/BIRC5 and macrophage infiltration (CD68, MAC387, CLEVER-1) on the basis of molecular subtypes of bladder cancer (KRT5, KRT20, ERBB2) to predict outcome of adjuvant treated muscle-invasive bladder cancer patients with regard to progression-free survival (PFS) and disease-specific survival (DSS).

However, there are some drawbacks in study design. Authors used primary tumor tissues before cystectomy including NMIBC. That doesn't make any sense. For evaluating the prognostic markers of MIBC after adjuvant chemotherapy, tissues from cystectomy should be utilized. Moreover, only 39 cases (PCR) and 28 cases (IHC) were enrolled in this study. The number of study cohort was too small to conclude any reliable results. Authors should analyze the expression level of each marker according to clinicopathological findings. Many corrections are needed for this paper to be accepted.

Author Response

Thanks a lot for that helpful comment. Our study’s main aim was to evaluate the role of validated markers in the special situation of adjuvant chemotherapy for muscle-invasive bladder cancer. We agree that our cohort is relatively small. However, the cohort of our study is quite homogenous regarding the treatment options. Comparing with other important studies in that context the number of patients n=50 is not too small. As one example, the discovery cohort to first describe the luminal and basal subtype in bladder cancer and to describe their responsiveness to neoadjuvant chemotherapy was based on 73 patients from MDCC (Choi et al. Cancer Cell 2014). By the way that cohort has been included in big data calculations predicting clinical markers (Galsky M et al. Cancer 2013; Haines L et al. Clin Genitourin Cancer 2013) because of its strict documentation of clinical parameters and follow-up. However, the findings of that study are showing significant results on RNA as well as protein based validation of the markers to prove the relvancve by independent technologies.

In many studies before it could be shown that diagnostics and prediction based on primary tumor tissue is important (Breyer J et al. Virch Arch 2017). The main goal of such studies lies in the fact to aid treatment decisions based on findings in primary tumor tissue. In former studies it could be shown that molecular characteristics of TURB and cystectomy material is very similar. We have had the possibility to demonstrate this for some cases from this study.

The findings of our study are limited by the small patient group and its retrospective character, but based on its results and in context of the previous larger studies using identical assay systems we demonstrate that further prospective studies with a higher number of patients are worth pursuing. We made some changes in our manuscript to point out these aspects.

Reviewer 3 Report

The Manuscript submitted by Ecke et al. reports about the prognostic role of Survivin (BIRC5) and markers for macrophage infiltration (CD68, MAC387, CLEVER-1) in muscle invasive bladder cancer (MIBC). They analysed protein expression of mentioned markers by immunohistochemistry (IHC) in a small cohort of patients (n=50, but several patients drop out due to lacking data on protein or RNA expression and 18 patients were <pT2) with adjuvant chemotherapy. Survivin expression was analysed for cytoplasmic and nuclear expression.

RNA Expression of BIRC5 and CD68 was also investigated by RT-qPCR. In addition, the authors measured RNA expression of KRT5, KRT20 and ERBB2 which are components of a marker panel for molecular subtyping. By correlation with patients´clinical data the authors found that high levels of nuclear Survivin staining was associated with better PFS and DSS. This predictive value was further improved by expression analysis of KRT20.

Macrophage infiltration was dominantly found in basal subtypes (which were only identified by analysis of KRT5, KRT20 and ERBB2) and associated with worse DSS in BIRC5 and ERBB2 double positive tumors. The authors mention that results were obtained by three independent centers, but do not further explain protocols, variation in results etc.

The study is adding information to the ongoing discussion on the best suitable analyte (RNA, protein, serum, urine) for the development of prognostic biomarkers or marker panels for molecular subtyping of bladder cancer. Unfortunately, the study design is not always sound.

Major points:

  1. The study cohort is fairly small,l even though 17 authors from three centers (six different sites and one company) were involved. The authors state that the focus of the manuscript is on MIBC. However, of the 50 patients included between 1996 and 2006, 18 patients have a T stage <pT2, so they were not MIBC. Patients were said to have progressed later. In addition, the data sets for protein and RNA analysis are not complete for all 50 patients. The PCR cohort included 39 cases. Since the manuscript wanted to compare performance of RNA and protein analysis or determine the predictive value of combined analysis, those cases are highly interesting with both PCR and IHC data. This was available in only 28 cases. I wonder why it was not possible to collect more and recent samples from the involved centers? This limits statistical significance or lacking significance as shown in fig. 3 and further correlation analyses.
  2. I assume that the authors investigated samples collected during a clinical study, since the adjuvant treatment scheme is not standard? Usually BC patients do not receive Paclitaxel. Thus, it has to be questioned to which extend results from this cohort can be transferred to other adjuvant treated cohorts, that usually do not receive paclitaxel? I also assume that this might be the reason why no validation cohort was investigated? However, to answer my question on applicability to other cohorts with rather standard treatment of care, the authors may have access to further patient cohorts, that could be analysed. This would increase soundness of the data and they would become more representative for MIBC patients. In conclusion, I doubt that this study proves that Survivin may be a good predictive marker for chemotherapy benefit (p14, line 314).
  3. I highly recommend to describe much more detailed the analysis procedure that must have been shared between the three independent centers. Also the data obtained by the three centers should be presented more clearly and experimentator or device variation should be discussed. In the current version it does not become clear whether the data shown represents the mean of all data from the three centers or just one representative dataset? Generally, the results could be visualized more clear and nice. For example, Table 3a/b have as parameters “Green vs Red” ect, which can only be understood by looking again at the previous analyses. In addition, the whole manuscript appears to have been written “a bit in a rush”. For example, chapter 2.2 and 2.3 have the same headline and content. Please double check phrasing and grammar. Use “.” and not “,” when listing p-values and other quantitative measures. I also recommend to use consistently Survivin when describing protein analysis and BIRC5 in case of RNA data.
  4. I disagree with the authors that it is common sense to measure only KRT5 and KRT20 as hallmarks for discrimination between the luminal and basal molecular subtypes. Associations with histologic variants and prognosis have been published previously. To my understanding the minimal marker set in PCR analysis for molecular subtyping, particularly of MIBC, is an ongoing discussion/matter of research. Usually, other additional markers, also more specific for basal types are measured. Thus, I recommend an extended analysis of markers to correlate the subtype with macrophage infiltration. Particularly since KRT5 and KRT20 mRNA could only be detected in 31 of 39 and 24 of 39 samples, respectively. The same applies to BIRC5. In addition, three markers of macrophage infiltration were measured by protein analysis, but KRT20 only by RNA analysis. Why did the authors not stain KRT20 simultaneously by IHC? Since a tissue microarray (TMA) was used, this would not have been difficult to do and rather more straight forward. The authors argue that analysis of macrophage infiltration is difficult by IHC, so that RNA quantification would be advantageous. However, then I miss a detailed description and discussion of the obtained IHC data with the three different macrophage markers showing quite different distribution/abundance to conclude that sole analysis of CD68 on the RNA level could be sufficient.
  5. Why was association between RNA expression and clinical parameters like PS, age, gender, T stage and old grading calculated by Pearson correlation and not by other statistical analyses?
  6. Usually a study number of ethics consent needs to be provided.

Author Response

The Manuscript submitted by Ecke et al. reports about the prognostic role of Survivin (BIRC5) and markers for macrophage infiltration (CD68, MAC387, CLEVER-1) in muscle invasive bladder cancer (MIBC). They analysed protein expression of mentioned markers by immunohistochemistry (IHC) in a small cohort of patients (n=50, but several patients drop out due to lacking data on protein or RNA expression and 18 patients were <pT2) with adjuvant chemotherapy. Survivin expression was analysed for cytoplasmic and nuclear expression.

RNA Expression of BIRC5 and CD68 was also investigated by RT-qPCR. In addition, the authors measured RNA expression of KRT5, KRT20 and ERBB2 which are components of a marker panel for molecular subtyping. By correlation with patients´clinical data the authors found that high levels of nuclear Survivin staining was associated with better PFS and DSS. This predictive value was further improved by expression analysis of KRT20.

Macrophage infiltration was dominantly found in basal subtypes (which were only identified by analysis of KRT5, KRT20 and ERBB2) and associated with worse DSS in BIRC5 and ERBB2 double positive tumors. The authors mention that results were obtained by three independent centers, but do not further explain protocols, variation in results etc.

The study is adding information to the ongoing discussion on the best suitable analyte (RNA, protein, serum, urine) for the development of prognostic biomarkers or marker panels for molecular subtyping of bladder cancer. Unfortunately, the study design is not always sound.

Major points:

1.The study cohort is fairly small,l even though 17 authors from three centers (six different sites and one company) were involved. The authors state that the focus of the manuscript is on MIBC. However, of the 50 patients included between 1996 and 2006, 18 patients have a T stage <pT2, so they were not MIBC. Patients were said to have progressed later. In addition, the data sets for protein and RNA analysis are not complete for all 50 patients. The PCR cohort included 39 cases. Since the manuscript wanted to compare performance of RNA and protein analysis or determine the predictive value of combined analysis, those cases are highly interesting with both PCR and IHC data. This was available in only 28 cases. I wonder why it was not possible to collect more and recent samples from the involved centers? This limits statistical significance or lacking significance as shown in fig. 3 and further correlation analyses.

Answer: The reviewer addresses some questions regarding the number of patients and involved centers. To clarify our approach we made some changes in the manuscript. The cohort of patients came only from one site. The used material from FFPE tumor tissue was analysed in three different centers. Due to the long duration of follow-up some members of the working group had already moved to other institutions.

2.I assume that the authors investigated samples collected during a clinical study, since the adjuvant treatment scheme is not standard? Usually BC patients do not receive Paclitaxel. Thus, it has to be questioned to which extend results from this cohort can be transferred to other adjuvant treated cohorts, that usually do not receive paclitaxel? I also assume that this might be the reason why no validation cohort was investigated? However, to answer my question on applicability to other cohorts with rather standard treatment of care, the authors may have access to further patient cohorts, that could be analysed. This would increase soundness of the data and they would become more representative for MIBC patients. In conclusion, I doubt that this study proves that Survivin may be a good predictive marker for chemotherapy benefit (p14, line 314).

Answer: The reviewer is right, that triple chemotherapy is not yet standard treatment. However, the chemotherapy regimen has shown comparable clinical efficacy comparable to other triple combinations and  to standard regimen as published (Galsky et al Cancer 2013). The addition of paclitaxel to the cisplatin-based chemotherapy backbone implies additional anti-tumor activity particularly against proliferative cancer tissues. In view of this our finding that tumors exhibiting low proliferative activity as shown by immunohistochemical of survivin and molecular assessment of its mRNA level is of even greater importance. It shows in a very homogeneously treated cohort, that even intensified treatment regimen cannot overcome the lack of sensitivity of tumors of low proliferative activity while indicating that this aspect can be generalized also for cisplatin based regimen lacking taxol treatment. We have included this aspect in the discussion of the results.

3.I highly recommend to describe much more detailed the analysis procedure that must have been shared between the three independent centers. Also the data obtained by the three centers should be presented more clearly and experimentator or device variation should be discussed. In the current version it does not become clear whether the data shown represents the mean of all data from the three centers or just one representative dataset? Generally, the results could be visualized more clear and nice. For example, Table 3a/b have as parameters “Green vs Red” ect, which can only be understood by looking again at the previous analyses. In addition, the whole manuscript appears to have been written “a bit in a rush”. For example, chapter 2.2 and 2.3 have the same headline and content. Please double check phrasing and grammar. Use “.” and not “,” when listing p-values and other quantitative measures. I also recommend to use consistently Survivin when describing protein analysis and BIRC5 in case of RNA data.

Answer: We agree with the reviewer, that these aspects have to be very clear and have added passages to clarify that the clinical data set stem form one center, where all patients had been treated in a consecutive manner, while the immunohistochemical and molecular data were generated at three specialized diagnostic centers. We have renamed table 3a/b to be more precise on the molecular signature type of the different groups. We have renamed chapter 2.2. and 2.3,  replaced “,” by “.” and corrected the distinction using “Survivin” or “BIRC5”, when describing protein or mRNA results respectively as correctly indicated by the reviewer

4.I disagree with the authors that it is common sense to measure only KRT5 and KRT20 as hallmarks for discrimination between the luminal and basal molecular subtypes. Associations with histologic variants and prognosis have been published previously. To my understanding the minimal marker set in PCR analysis for molecular subtyping, particularly of MIBC, is an ongoing discussion/matter of research. Usually, other additional markers, also more specific for basal types are measured. Thus, I recommend an extended analysis of markers to correlate the subtype with macrophage infiltration. Particularly since KRT5 and KRT20 mRNA could only be detected in 31 of 39 and 24 of 39 samples, respectively. The same applies to BIRC5. In addition, three markers of macrophage infiltration were measured by protein analysis, but KRT20 only by RNA analysis. Why did the authors not stain KRT20 simultaneously by IHC? Since a tissue microarray (TMA) was used, this would not have been difficult to do and rather more straight forward. The authors argue that analysis of macrophage infiltration is difficult by IHC, so that RNA quantification would be advantageous. However, then I miss a detailed description and discussion of the obtained IHC data with the three different macrophage markers showing quite different distribution/abundance to conclude that sole analysis of CD68 on the RNA level could be sufficient.

Answer: We agree with the reviewer that  the minimal marker set for molecular subtyping by PCR or immunohistochemistry is an ongoing matter of debate and research. We have adopted the respective passage by softening the phrase and avoiding the term “hallmark”. However, when looking at the initial and subsequently  the overexpression of KRT20 in luminal tumors and of KRT5 in basal tumors have been integral part of all molecular signatures being applied, while the exact algorithms and number of subtypes is in the process of consolidation. As one example the initial hypothesis generating distinction of muscle invasive bladder cancer  (n=73)  into basal and luminal tumors by Choi et al (Cancer Cell 2014) to predict response to neoadjuvant  cisplatin based chemotherapy was based on upregulated genes containing signature biomarkers for basal (CD44, KRT5, KRT6, KRT14, and CDH3) and luminal (CD24, FOXA1, GATA3, ERBB2, ERBB3, XBP1, and KRT20) subtypeswhile referencing to landmark papers of breast cancers, respectively. Furthermore within this first description of basal and luminal subtypes in bladder cancer the subtyping was recapitulated by IHC of only CK5/6, CD44 and CK20 on protein basis and KRT5, KRT6, CD44 as well as FOXA1 and KRT20 on mRNA basis (see figure 1 “Basal and luminal subtypes of bladder cancer”, Choi et al. Cancer Cell 2014) indicating equivalence of mRNA based and IHC-based distinction. In subsequent own previous  studies of larger retrospective cohorts, we have validated the prognostic value of assessing only KRT5 and KRT20 for molecular subtyping of NMIBC and MIBC (Breyer et al. Virchows Arch 2017; Eckstein et al. Int J Mol Sc. 2018) and furthermore compared to histological subtypes of MIBC, thereby confirming the applicability of KRT5 and KRT20 as minimal marker set. Unfortunately, the tissue availability of the current cohort consisting of tissue microarray dots is quite limited. Therefore we had to focus on less markers. In addition, fresh cuts for additional IHC assessments were not available. We agree that this would have been fortunate and straight forward. However, in previous trials we have already shown good comparability of mRNA and protein-based Keratin assessment (Breyer et al Virchows Arch 2017), which is also in line with the initial publication of Choi et al. The extension of the marker set and correlation of a multitude of subtyping markers with macrophage markers  is of course interesting and being planned for the upcoming prospective trial. The current study is apilot study that has raised substantial results indicating that the role of macrophage markers might be different from T-cell based immune assessment and therefore justifies further efforts for in depth analysis in larger cohorts. In this context and in line with the comments of the reviewer we do not draw the conclusion that sole determination of CD68 would be sufficient to assess the relevance of macrophage infiltration, but rather show that macrophages do have prognostic value both on protein and mRNA level, which warrants further more detailed analysis on subsequent, larger cohorts. Still we find the initial finding as being of general value in view of the fact that most immune assessments focus frequently only on T-cell infiltrates and therefore included these aspects in the current manuscript.

5.Why was association between RNA expression and clinical parameters like PS, age, gender, T stage and old grading calculated by Pearson correlation and not by other statistical analyses?

Answer: In that point the reviewer’s comment is right. But we have chosen overview display of Pearsons correlation for better visibility with similar statistical results being found by other statistical analyses such as Spearman correlation.

6.Usually a study number of ethics consent needs to be provided.

Answer: Thanks a lot for that advice. We have included the necessary details in the Material and Methods section.

Reviewer 4 Report

  1. The authors make a great effort to identify the potential markers and influence factors in MIBC.
  2. Please answer that the specimens are collected during 1996-2006 or 2001-2006?
  3. Why used transurethral resected specimens? Why not radical cystectomy specimens?
  4. For those whose disease were progressed from NMIBC to MIBC and compared to those initial presented with MIBC or metastatic disease, any remarkable difference in markers expression?
  5. The authors should condense and focus to facilitate the reading availability of this article.
  6. Some of the typewriting errors need to correct.

Author Response

1.The authors make a great effort to identify the potential markers and influence factors in MIBC.

Answer: Thanks a lot for that friendly comment. We agree and appreciate the comment.

2.Please answer that the specimens are collected during 1996-2006 or 2001-2006?

Answer: The reviewer is right. Of course, we corrected the wrong data of sample collection in the figure 1.

3.Why used transurethral resected specimens? Why not radical cystectomy specimens?

Answer: In many studies before it could be shown that diagnostics and prediction based on primary tumor tissue is important (Breyer J et al. Virch Arch 2017). The main goal of such studies lies in the fact to find and develop treatment decisions based on tissue analysis of primary tumor tissue. In former studies it could be shown that molecular characteristics of TURB and cystectomy material is very similar. Moreover the initial introduction of molecular  subtyping of muscle invasive bladder cancer has been done on TUR biopsies (Choi et al Cancer Cell 2014). This is in line with the reasoning to predict outcome of neodjuvant treatment and to evaluate potentially predictive marker algorithms that could help to decide whether to do neoadjuvant or adjuvant chemotherapy.

4.For those whose disease were progressed from NMIBC to MIBC and compared to those initial presented with MIBC or metastatic disease, any remarkable difference in markers expression?

Answer: Comparison of NMIBC and MIBC was possible for only 4 patients. Marker gene expression was comparable. However, with regard to KRT20 expression one  MIBC did exhibit significantly increased expression of the luminal marker KRT20. The reviewer comment is important and we included this side result in the manuscript. Unfortunately the number of results regarding that question is too low to make any statistical calculations.

5.The authors should condense and focus to facilitate the reading availability of this article.

Answer: Thanks a lot for that helpful comment. We made some changes in the manuscript to improve readability of the article.

6.Some of the typewriting errors need to correct.

Answer: All authors have carefully read and edited the revised manuscript for typewriting errors.

Round 2

Reviewer 1 Report

I am satisfied with the authors response to my comments.

Author Response

Thanks a lot for your review.

We are happy to have the possibility ameliorate our manuscript.

Reviewer 3 Report

no comment

Author Response

Thanks a lot for the review of our manuscript.

Due to your helpful comments we could ameliorate the paper.

Reviewer 4 Report

  1. The authors revised the article as suggestions. There are two minor issues need to answered.
  2. In page 3, you showed that "the total study cohort consisted of 50 MIBC tumor patients treated from 1996 to 2006 at a single institute". This is different from Figure 1 in page 4 that stated the cases were collected 2001 to 2006. Please make sure which period is corrected.
  3. In page 1 "ABSTRACT" section,  you mentioned that "all patients had been treated radical cystectomy followed by adjuvant triple chemotherapy". If this is the situation of tissue specimens you held, then why don't you use those specimens from radical cystectomy but instead of to use those from TUR-biopsy specimens? The molecular biological characteristics of these two types of specimens obtained may have some different results. What's your opinion regarding this? 

Author Response

The authors revised the article as suggestions. There are two minor issues need to answered.

In page 3, you showed that "the total study cohort consisted of 50 MIBC tumor patients treated from 1996 to 2006 at a single institute". This is different from Figure 1 in page 4 that stated the cases were collected 2001 to 2006. Please make sure which period is corrected.

Answer: Thanks again for that comment. We agree that this information is a little bit confusing. The treatment of these patients starting with cystectomy was from 2001 to 2006, but the date of diagnosis was in some cases earlier. We changed that in the manuscript from „treated“ to „diagnosed“.

In page 1 "ABSTRACT" section,  you mentioned that "all patients had been treated radical cystectomy followed by adjuvant triple chemotherapy". If this is the situation of tissue specimens you held, then why don't you use those specimens from radical cystectomy but instead of to use those from TUR-biopsy specimens? The molecular biological characteristics of these two types of specimens obtained may have some different results. What's your opinion regarding this?

Answer: This is good point. Actually we have discussed this intensively in the study design phase. Our vision is to predict responsiveness of tumors based on initial TUR biopsies. Therfore we have done extensive analysis of pT1 and (non chemotherapy treated) MIBC series based on TUR biospy analysis (Breyer et al. Virchows Arch 2017; Eckstein et al. IJMS 2018) and are interested to compare the data between the series. Moreover, one of our focus is to predict response to neoadjuvant chemotherapy based on TUR biopsy assessment as recently published at ASCO GU (Roghmann et al. Abstract No. 562, publication on preparation). The current analysis was intended to compare with these TUR biopsy based studies, therefore we have decided to focus on TUR biospies as the results may contribute to future decision making wether to apply (neo)adjuvant chemotherapy, stay with adjuvant treatment or include in upcoming targeted therapy regimen trials. However, as mentioned in the mansucript this study is a pilot study and shall lay the basis for prospective validation series. As part of these trials it is planned to extensive compare TURB and cystectomy samples of adjuvant and neoadjuvantly treated patients. However, in view of initial decision making and potential neoadjuvant options we regard TUR biopsy analysis to be of particular importance, which has been the reason for the intial study design.

Round 3

Reviewer 4 Report

The authors revised this article in proper way.

Author Response

(The authors gave the same response as above.)
